# FEASIBLE ADVERSARIAL ROBUST REINFORCEMENT LEARNING FOR UNDERSPECIFIED ENVIRONMENTS

## ABSTRACT

Robust reinforcement learning (RL) considers the problem of learning policies that perform well in the worst case among a set of possible environment parameter settings. In real-world environments, choosing the set of allowed parameter settings for robust RL can be a difficult task. When that set is specified too narrowly, the agent will be left vulnerable to reasonable parameter values unaccounted for. When specified too broadly, the agent will be too cautious. In this paper, we propose Feasible Adversarial Robust RL (FARR), a novel problem formulation and objective for automatically determining the set of environment parameter values over which to be robust. FARR implicitly defines the set of feasible parameter values as those on which an agent could achieve a benchmark reward given enough training resources. By formulating this problem as a two-player zero-sum game, optimizing the FARR objective jointly produces an adversarial distribution over parameter values with feasible support and a policy robust over this feasible parameter set. We demonstrate that approximate Nash equilibria for this objective can be found using a variation of the PSRO algorithm. Furthermore, we show that an optimal agent trained with FARR is more robust to feasible adversarial parameter selection than with existing minimax, domain-randomization, and regret objectives in a parameterized gridworld and three MuJoCo control environments.

## 1 INTRODUCTION

Recent advancements in deep reinforcement learning (RL) show promise for the field's applicability to control in real-world environments by training in simulation (OpenAI et al., 2018; Hu et al., 2021; Li et al., 2021). In such deployment scenarios, details of the test-time environment layout and dynamics can differ from what may be experienced at training time. It is important to account for these potential variations to achieve sufficient generalization and test-time performance.

Robust RL methods train on an adversarial distribution of difficult environment variations to attempt to maximize worst-case performance at test-time. This process can be formulated as a two-player zero-sum game between the primary learning agent, the *protagonist*, which is a maximizer of its environment reward, and a second agent, the *adversary*, which alters and affects the environment to minimize the protagonist's reward (Pinto et al., 2017). By finding a Nash equilibrium in this game, the protagonist maximizes its worst-case performance over the set of realizable environment variations.

This work aims to address the growing challenge of specifying the robust adversarial RL formulation in complex domains. On the one hand, the protagonist's worst-case performance guarantee only applies to the set of environment variations realizable by the adversary. It is therefore desirable to allow an adversary to represent a large uncertainty set of environment variations. On the other hand, care has to be taken to prevent the adversary from providing unrealistically difficult conditions. If an adversary can pose an insurmountable problem in the protagonist's training curriculum, under a standard robust RL objective, it will learn to do so, and the protagonist will exhibit overly cautious behavior at test-time (Ma et al., 2018). As the complexity of environment variations representable in simulation increases, the logistic difficulty of well-specifying the limits of the adversary's abilities is exacerbated.

For example, consider an RL agent for a warehouse robot trained to accomplish object manipulation and retrieval tasks in simulation. It is conceivable that the developer cannot accurately specify

the distribution of environment layouts, events, and object properties that the robot can expect to encounter after deployment. A robust RL approach may be well suited for this scenario to, during training, highlight difficult and edge-case variations. However, given the large and complex uncertainty set of environment variations, it is likely that an adversary would be able to find and over-represent unrealistically difficult conditions such as unreachable item locations or a critical hallway perpetually blocked by traffic. We likely have no need to maximize our performance lower bound on tasks harder than those we believe will be seen in deployment. However, with an increasingly complex simulation, it may become impractical to hand-design rules to define which variation is and is not too difficult.

To avoid the challenge of precisely tuning the uncertainty set of variations that the adversary can and cannot specify, we consider a new modified robust RL problem setting. In this setting, we define an environment variation provided by an adversary as *feasible* if there exists any policy that can achieve at least $\lambda$ return (i.e. policy success score) under it and *infeasible* otherwise. The parameter $\lambda$ describes the level of worse-case test-time return for which we wish our agent to train. Given an underspecified environment, i.e. an environment parameterized by an uncertainty set which can include unrealistically difficult, infeasible conditions, our goal is to find a protagonist robust only to the space of feasible environment variations.

With this new problem setting, we propose Feasible Adversarial Robust Reinforcement learning (FARR), in which a protagonist is trained to be robust to the space of all feasible environment variations by applying a reward penalty to the adversary when it provides infeasible conditions. This penalty is incorporated into a new objective, which we formulate as a two-player zero-sum game. Notably, this formulation does not require a priori knowledge of which variations are feasible or infeasible.

We compare a near-optimal solution for FARR against that of a standard robust RL minimax game formulation, domain randomization, and an adversarial regret objective similarly designed to avoid unsolvable tasks (Dennis et al., 2020; Parker-Holder et al., 2022). For the two-player zero-sum game objectives of FARR, minimax, and regret, we approximate Nash equilibria using a variation of the Policy Space Response Oracles (PSRO) algorithm (Lanctot et al., 2017). We evalaute in a gridworld and three MuJoCo environments. Given underspecified environments where infeasible variations can be selected by the adversary, we demonstrate that FARR produces a protagonist policy more robust to the set of feasible task variations than existing robust RL minimax, domain-randomization, and regret-based objectives. To summarize, the primary contributions of this work are:

- We introduce FARR, a novel objective designed to produce an agent that is robust to the feasible set of tasks implicitly defined by a threshold on achievable reward.

- We show that this FARR objective can be effectively optimized using a variation of the PSRO algorithm.

- We empirically validate that a near-optimal solution to the FARR objective results in higher worst-case reward among feasible tasks than solutions for other objectives designed for similar purposes: standard robust RL minimax, domain randomization, and regret, in a parameterized gridworld and three MuJoCo environments.

## 2 RELATED WORK

### 2.1 DOMAIN RANDOMIZATION

Domain randomization methods train in simulation on a distribution of environment variations that is believed to generalize to the real environment. The choice of a distribution for training-time simulation parameters plays a major role in the final test performance of deployed agents (Vuong et al., 2019). While domain randomization has been used with much success in sim-to-real settings (OpenAI et al., 2018; Tobin et al., 2017; Hu et al., 2021; Li et al., 2021), its objective is typically the average-case return over the simulation uncertainty set rather than the worst-case. This can be desirable in applications where average training-time performance is known to generalize to the test environment or where this can be validated. However, average-case optimization can be unacceptable in many real-world scenarios where safety, mission-criticality, or regulation require

the agent to perform well in test environments whose weight in the average would be small and where sufficient validation is unavailable.

## 2.2 Robust Reinforcement Learning

Robust reinforcement learning methods optimize reward in the worst-case and provide a promising approach for sim-to-real. Robust Adversarial Reinforcement Learning (RARL) (Pinto et al., 2017), which this work builds upon, optimizes return under the worst-case environment variations by optimizing a two-player zero-sum game between a task-performing protagonist and an environment-controlling adversary. Numerous mechanisms by which the adversary affects the environment have been explored including perturbing forces and varying environment dynamics (Pinto et al., 2017; Mandlekar et al., 2017; Nakao et al., 2021), disturbances to actions (Tessler et al., 2019; Vinitsky et al., 2020; Tan et al., 2020), and attacks on agent observations (Pattanaik et al., 2017; Gleave et al., 2019; Zhang et al., 2020; 2021; Kumar et al., 2021). However, each of these works makes the assumption that the uncertainty set from which perturbations and variations may be sampled is well-specified and tuned such that an agent robust to a minimax selection over the entire set will perform optimally in deployment. FARR relaxes this assumption and intends to produce a robust agent to the real environment, even when the adversary can select unrealistically hard variations.

## 2.3 Automatic Curriculum Design

Similar to finding the worst-case distribution of environment variations is finding the best-case distribution for improving an existing agent. Automatic curricula seek to find environment parameters that are challenging but not too difficult for a training agent. Racaniere et al. (2019), Campero et al. (2020), and Florensa et al. (2018) generate useful curricula for a single learning agent to solve difficult tasks by proposing appropriately challenging goals for the agent's current abilities. POET (Wang et al., 2019; 2020) co-evolves a population of tasks and associated agents to produce an increasingly complex set of tasks with coupled agents capable of solving their associated task. While each of these methods can create increasingly capable agents, they offer no guarantees for final agent robustness.

Asymmetric self-play (Sukhbaatar et al., 2018; OpenAI et al., 2021) facilitates two agents with similar capabilities to compete in a two-player game where one attempts to reach goals that are achievable but difficult to the other agent. PAIRED (Dennis et al., 2020) and subsequent improvements to optimization (Parker-Holder et al., 2022; Du et al., 2022) extend this concept beyond goals to the generation of parameterized environments by introducing a two-player zero-sum regret objective between an adversary and a protagonist. The adversary learns to specify environment conditions in which regret is highest such that the protagonist's performance most differs from estimated optimal performance. Regret selects tasks where changes to protagonist behavior could most affect incurred reward and is effective for training a broadly capable agent given limited preference over tasks. Regret can also ensure successful performance where possible in certain classes of environments with well-behaved reward functions. However, because the protagonist can have imperfect information on the current task and because a Nash equilibrium regret distribution does not minimize protagonist reward over a target set of tasks, a regret protagonist does not maximize worst-case performance over the feasible set of tasks like we are interested in with FARR.

## 3 Background

### 3.1 Underspecified Partially-Observable Markov Decision Processes

We adapt Underspecified Partially-Observable Markov Decision Processes (UPOMDPs) from Dennis et al. (2020) where there exists a parameter of variation $\theta \in \Theta$ that is hidden from the agent. This parameter $\theta$ controls some aspect of the environment that can potentially be discovered through interaction. For example, $\theta$ could control the friction of a robot arm, the mass of a cartpole, or the layout of a room.

We model a UPOMDP as a tuple $\mathcal{M} = \langle A, S, O, \Theta, \mathcal{T}, \rho, \mathcal{I}, \mathcal{R}, \gamma \rangle$ where $A$ is the set of actions, $S$ is the set of states, $O$ is the set of observations, and $\gamma$ is the discount factor. The choice of parameter $\theta \in \Theta$ controls the conditions in this environment, namely the transition distribution

function $\mathcal{T} : S \times A \times \Theta \to \Delta(S)$, the initial state distribution $\rho : \Theta \to \Delta(S)$, the observation function $\mathcal{I} : S \times \Theta \to O$, and the reward function $\mathcal{R} : S \times \Theta \to \mathbb{R}$.

Given an observable history $h \in H = (O \times A)^* \times O$, a protagonist policy $\pi_p : H \to \Delta(A)$ decides at each time step $t$ on the distribution of the action $a_t$ after seeing $h = o_0, a_0, \ldots, o_t$. Jointly with a UPOMDP $\mathcal{M}$ and a specific environment parameter $\theta$, the policy induces a distribution $p_{\pi_p}^{\theta}$ over the states, actions, observations, and rewards in an interaction episode. We define the protagonist's utility $U_p(\pi_p, \theta) = \mathbb{E}_{p_{\pi_p}^{\theta}} \left[ \sum_t \gamma^t r_t \right]$ as the expected episode discounted return for a protagonist policy $\pi_p$ and choice of $\theta$.

## 3.2 POLICY SPACE RESPONSE ORACLES

Policy Space Response Oracles (PSRO) (Lanctot et al., 2017) is a deep RL method for calculating approximate Nash equilibria (NE) in zero-sum two-player games. It extends the normal-form Double-Oracle algorithm (McMahan et al., 2003) to games with sequential interaction. At a high-level, PSRO iteratively adds new policies for each player to a population until a normal-form mixed-strategy solution to the restricted game induced by selecting population policies closely approximates a NE in the full game.

PSRO operates by maintaining a population of policies $\Pi_i$ for each player $i$ and a normal-form mixed strategy $\sigma_i \in \Delta(\Pi_i)$. This mixed strategy $\sigma_i$ represents a distribution over policies $\pi_i \in \Pi_i$ to sample from at the beginning of each episode, and upon algorithm termination, $\sigma = (\sigma_1, \sigma_2)$ is the final output of PSRO. The utility of player $i$ of playing a mixed strategy $\sigma_i$ against an opponent's policy $\pi_{-i}$ is therefore $U_i(\sigma_i, \pi_{-i}) = \mathbb{E}_{\pi_i \sim \sigma_i} [U_i(\pi_i, \pi_{-i})]$, and likewise, $U_i(\sigma_i, \sigma_{-i}) = \mathbb{E}_{\pi_i \sim \sigma_i, \pi_{-i} \sim \sigma_{-i}} [U_i(\pi_i, \pi_{-i})]$.

In each iteration of PSRO, new policies for each player $i$ are added to its population $\Pi_i$. In the case of sequential interaction, this is typically an RL best-response $\mathbb{BR}(\sigma_{-i}) = \arg\max_{\pi_i} U_i(\pi_i, \sigma_{-i})$ that maximally exploits the opponent mixed-strategy. However, this choice of new policy is not a requirement, and PSRO maintains NE convergence guarantees so long as novel policies are continuously added for each player.

After adding new policies, utilities $U^{\Pi}(\pi_1, \pi_2)$ between each pairing of player policies $\pi_1 \in \Pi_1$ and $\pi_2 \in \Pi_2$ are empirically estimated using rollouts to create a normal-form restricted game in which population policies are the strategies. A new NE mixed-strategy that solves the restricted game, a *restricted NE* $\sigma = (\sigma_1, \sigma_2)$, is then cheaply calculated for each player. This process is repeated until no new policies are added to either player's population or the process is externally stopped. As the number of strategies in each player's population grows, a NE solution to the restricted game asymptotically converges to a NE solution to the full game.

While other potentially suitable methods for solving extensive-form games exist, for example NFSP (Heinrich & Silver, 2016) and Deep-CFR (Brown et al., 2019), PSRO was chosen out of practicality because it can be implemented as an additional logic layer on top of existing reinforcement learning software stacks. Although PSRO-based methods are competitive in sample-efficiently solving two-player zero-sum games (Lanctot et al., 2017; Vinyals et al., 2019; McAleer et al., 2021; 2022b;a; Liu et al., 2022), the purpose of experiments in this work is not focused on improving the speed at which we might reach optima. Rather, we are interested in ensuring that we can reliably reach approximate NE for each objective we test by using PSRO in order to compare their near-optimal solutions on even ground.

## 4 FEASIBLE ADVERSARIAL ROBUST REINFORCEMENT LEARNING

### 4.1 MOTIVATION

We begin our discussion of the FARR method with a motivating example. Consider a cartpole environment where the pole is subject to perturbing forces of an unknown magnitude. To maximize our worse-case performance, we could formulate a robust RL game in which the adversary provides the most difficult possible mixed strategy of force magnitudes. By learning a best response strategy to this adversary, the protagonist will maximize its worst-case performance at test-time when presented with an unknown distribution of forces.

Importantly, to achieve this process, we would need to allow the adversary to specify a sufficiently wide range of force magnitudes such that the real environment is believed to be in this range. However, we would not want to allow the adversary to specify force magnitudes so large that the task becomes impossible, because the adversary would then always select overly difficult parameters and the protagonist would only train on impossible task variations, failing at test time.

In this cartpole example, it is possible to manually adjust the range of allowed force values until the widest possible uncertainty set is found that still avoids the learning of a degenerate protagonist strategy. However, if this setting were scaled up to specifying the allowed values of many coefficients, level layouts, or environmental events where the adversary has complex, high-dimensional interactions with the protagonist, hand tuning these limits may no longer be viable.

To remove the need for human expert tuning of the adversary limits, we instead create a game where the adversary is allowed to specify impossible environment variations but is heavily penalized for doing so, using the method we describe below.

## 4.2 FEASIBILITY

We define an environment parameter $\theta \in \Theta$ as $\lambda$-feasible if a best-response to $\theta$ can achieve an expected return of at least $\lambda \in \mathbb{R}$. Heuristically, the value for $\lambda$ could be set as the lowest average return that we would expect an agent to receive across variations in deployment if it could act optimally with respect to each variation. We define the feasible set $\mathcal{F}^\lambda$ of environment parameters as:

$$\mathcal{F}^\lambda = \{\theta \in \Theta | U_p(\mathbb{BR}(\theta), \theta) \geq \lambda\}. \tag{1}$$

$\mathcal{F}^\lambda$ matches our motivation for considering feasible parameters when either of two conditions is satisfied. First, we may have a lower bound on the achievable performance in the real environment, and we can set $\lambda$ at or below that bound. If the real environment is feasible, $\theta^* \in \mathcal{F}^\lambda$, then we are justified in avoiding training the protagonist on infeasible environments. Second, there may be a performance threshold such that only above it we have a preference over agent behaviors. For example, we may only care where a warehouse robot navigates if it has a valid path to its target shelf.

The set of feasible variations $\mathcal{F}^\lambda$ is generally unknown. As part of optimizing the FARR objective, the adversary will learn an approximation of $\mathcal{F}^\lambda$ and use it to guide the selection of a mixed strategy over discovered feasible environment variations. We note that, in order to measure robustness to $\mathcal{F}^\lambda$ in this paper, we focus on environments where we can, in fact, calculate $\mathcal{F}^\lambda$ for the purpose of test-time evaluation. It is reasonable to expect our findings to carry over to some domains where $\mathcal{F}^\lambda$ is truly unknown and where the method cannot be directly evaluated.

## 4.3 FARR OBJECTIVE

We wish to optimize the standard zero-sum robust adversarial game through PSRO with the additional constraint that the support of the adversary mixed strategy $\sigma_\theta$ contains only strategies in the feasible set $\mathcal{F}^\lambda$. Define $\mathrm{supp}(\sigma_\theta) = \{\theta \in \Theta | \sigma_\theta(\theta) > 0\}$. Our intended FARR objective is to solve the game:

$$\min_{\sigma_\theta} \max_{\sigma_p} U_p(\sigma_p, \sigma_\theta) \quad \text{subject to} \quad \mathrm{supp}(\sigma_\theta) \subseteq \mathcal{F}^\lambda. \tag{2}$$

An adversary mixed strategy $\sigma_\theta$ with support that is a subset of $\mathcal{F}^\lambda$ will only provide feasible environment variations to the protagonist.

In order to provide flexibility in how novel adversary strategies that satisfy this constraint could be optimized, we can replace the hard constraint with a sufficiently large penalty $C$ to the adversary when it violates the constraint. An equivalent zero-sum FARR objective would then be to optimize:

$$\min_{\sigma_\theta} \max_{\sigma_p} U_p^\lambda(\sigma_p, \sigma_\theta), \tag{3}$$

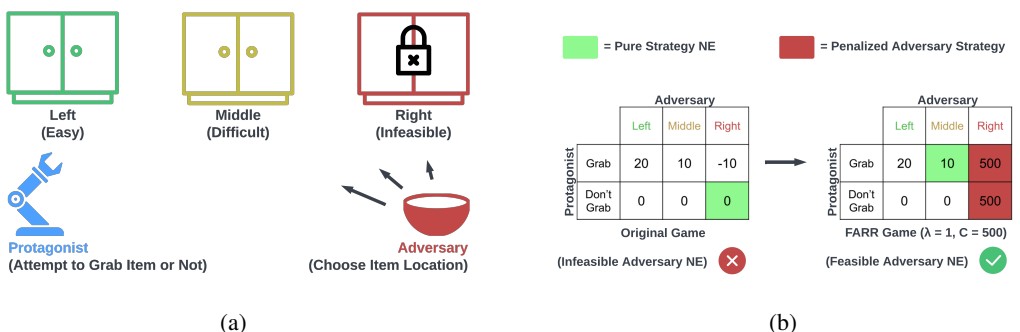

Figure 1: (a) An adversarial item retrieval game. The adversary can hide a bowl in a left, middle, or locked right cabinet. The protagonist chooses whether or not to attempt to find and grab the bowl. The protagonist receives a penalty for attempting to grab a bowl placed in the locked right cabinet and failing. (b) Matrix game representation of the original game and FARR transformed game. Protagonist utilities are shown. NE for the original game place weight on infeasible tasks with suboptimal behavior from the protagonist. NE for the FARR game provide feasible but difficult tasks and optimal protagonist behavior supposing feasible tasks are expected in deployment.

where the FARR utility function $U_p^\lambda(\pi_p, \theta)$ is unchanged from the original game if the adversary's provided environment parameter is feasible, and where otherwise a large constant adversary penalty $C$ is applied:

$$U_p^\lambda(\pi_p, \theta) = \begin{cases} C & \text{if } U_p(\mathbb{BR}(\theta), \theta) < \lambda \\ U_p(\pi_p, \theta) & \text{otherwise.} \end{cases} \tag{4}$$

## 4.4 MATRIX GAME EXAMPLE

We demonstrate the effect of the FARR utility function (equation 4) through a matrix game example shown in Figure 1. The adversary specifies the location of a bowl among three cabinets, where the middle cabinet is more difficult for the protagonist to access than the left, and the right cabinet is locked and inaccessible. Without observing the bowl's location, the protagonist must choose whether or not to attempt to find and grab the bowl, receiving a penalty for a failed retrieval attempt. In deployment, we expect that the item will always be placed in a feasible, accessible cabinet, either the left or the middle, so ideal behavior for the protagonist would be to always attempt to grab the bowl. For this game, we define the feasibility threshold over task variations as $\lambda = 1$.

The original game, shown in matrix-form in Figure 1(b), contains an infeasible adversary pure strategy (bowl in locked right cabinet). Because of the presence of this infeasible adversary strategy, the protagonist will never attempt to grab the bowl in any of the game's Nash equilibria (one pure strategy NE is displayed in green). If we now replace the original game's utility function $U_p$ with the FARR utility function $U_p^\lambda$ using $C = 500$, the adversary receives a penalty of $-500$ for placing the bowl in the infeasible locked right cabinet because there exists no protagonist strategy that can achieve a utility of a least $\lambda = 1$ under that environment variation. Instead, in the FARR transformed game, a NE adversary places the bowl in the feasible but difficult middle cabinet, and the protagonist learns, facing a selection of feasible tasks, to always attempt to retrieve the bowl. This new protagonist NE strategy for the FARR game is optimal with respect to anticipated feasible deployment conditions.

## 4.5 OPTIMIZING WITH PSRO

We use PSRO to solve for an approximate NE of the FARR transformed game with the penalty-based objective defined in equation (3). To optimize FARR with PSRO, only the scalar values $\lambda$ and $C$ need to be provided, where $\lambda$ is the maximum difficulty expected among tasks in deployment and $C$ is an arbitrarily large positive value. Our optimization process is shown in algorithm 1. We represent environment parameters $\theta$ as strategies in the adversary population $\Pi_\theta$ with an output mixed-strategy $\sigma_\theta$ over $\Pi_\theta$. Similarly, we represent protagonist RL agent policies as strategies $\pi_p \in \Pi_p$ with an

---

**Algorithm 1** FARR Optimized through PSRO

---

**Input:** $\lambda, C,$ and Initial policy sets $\Pi = (\Pi_p, \Pi_\theta)$ for Protagonist player and Adversary player
Compute expected FARR payoff matrix $U_\lambda^\Pi$ as utilities $U_p^\lambda(\pi_p, \theta)$ for each joint $(\pi_p, \theta) \in \Pi$
**repeat**
    Compute Normal-Form restricted NE $\sigma = (\sigma_p, \sigma_\theta)$ over population policies $\Pi$ using $U_\lambda^\Pi$
    Calculate new Protagonist policy $\pi_p$ (e.g. $\mathbb{BR}(\sigma_\theta)$)
    $\Pi_p = \Pi_p \cup \{\pi_p\}$
    **for** at least one iteration **do**
        Calculate new Adversary strategy $\theta$ and associated estimator for $\mathbb{BR}(\theta)$
        $\Pi_\theta = \Pi_\theta \cup \{\theta\}$
    **end for**
    Compute missing entries in $U_\lambda^\Pi$ from $\Pi$
**until** terminated early or no novel policies can be added
**Output:** current Protagonist restricted NE strategy $\sigma_p$

---

output mixed-strategy $\sigma_p$ over $\Pi_p$. In each PSRO iteration, to best-respond to the current adversary restricted NE $\sigma_\theta$, a new protagonist policy is trained using RL with a fixed environment experience budget. One or more random novel adversary pure strategies are added in each PSRO iteration. For wall-time parallelization, we add 3 in each iteration. For each adversary strategy $\theta$, we also train an evaluator RL policy $\pi_e^\theta$ with the same hyperpameters as the protagonist to estimate $\mathbb{BR}(\theta)$ and feasibility for the FARR utility function $U_p^\lambda$.

## 5 EXPERIMENTS

To illustrate the utility of filtering out infeasible tasks in a robust RL setting, we perform experiments in environments where overly difficult or unsolvable task variations are possible while measuring performance under feasible conditions. In a goal-based gridworld environment and three perturbed MuJoCo (Todorov et al., 2012) control environments, we compare the performance of FARR with three alternative objectives: a standard minimax robust adversarial RL objective, domain randomization, and the regret objective as proposed in Dennis et al. (2020). Given a threshold for feasibility $\lambda$, we evaluate worst-case performance within the set $\mathcal{F}^\lambda$ of feasible environment parameters. FARR outperforms each of these objectives because it is able to provide an adversarial training distribution of environment variations limited only to instances that are discovered to be feasible, while other methods provide overly difficult or otherwise mismatched training distributions for worst-case feasible conditions.

We optimize FARR and other baseline objectives with PSRO and identical protagonist RL best-response algorithms in order to compare final performance given guarantees of asymptotically reaching an approximate NE for each zero-sum objective.

We describe each of the baseline objectives below:

**Minimax** The standard objective for robust adversarial RL using unmodified $U_p(\sigma_p, \sigma_\theta)$. When infeasible tasks are allowed to the adversary, the standard minimax robust RL objective will focus on such tasks, resulting in overly cautious protagonist behavior or failed learning.

**Domain Randomization (DR)** With domain randomization, we train a single protagonist policy $\pi_p^{DR} = \mathbb{BR}(\sigma_\theta^{DR})$ to saturation against a uniform mixture of all possible environment variations $\sigma_\theta^{DR} = \mathcal{U}(\Theta)$. Domain randomization can result in an exploitable agent if the relevant feasible part of configuration space is underrepresented by the uniform measure. This objective is optimized by training a single RL policy rather than with PSRO.

**Regret** Matching the objective used by PAIRED (Dennis et al., 2020), we use PSRO to approximately solve for NE using the objective $\min_{\sigma_\theta} \max_{\sigma_p} \mathbb{E}_{\theta \sim \sigma_\theta} [U_p(\sigma_p, \theta) - U_p(\mathbb{BR}(\theta), \theta)]$. $\mathbb{BR}(\theta)$ is estimated using the same method as with FARR by training an evaluation agent $\pi_e^\theta$ against each $\theta$. While designed to provide a distribution of tasks where the protagonist is known to be able to positively affect its performance through optimal behavior, this curriculum learning objective does not generally provide a task distribution suitable for robust learning to a specific set of tasks such as $\mathcal{F}^\lambda$.

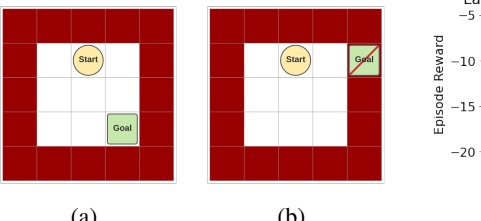
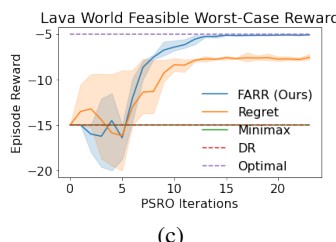

(a)    (b)    (c)

Figure 2: (a) The Lava World grid environment. The adversary specifies the location of an unobservable goal. The protagonist receives -1 reward each timestep until it reaches the hidden goal. If the protagonist steps in lava (red), the episode ends and it receives a penalty of -15 reward. (b) The environment is underspecified and the goal can be placed in lava, forcing the agent to receive the lava penalty and creating an infeasible task given $\lambda = -10$. (c) Solving for approximate NE using PSRO, the FARR objective results in an agent maximally robust to the feasible, non-lava goals while other objectives result in suboptimal worst-case performance among the feasible set of tasks.

To calculate the worst-case episode reward within the set $\mathcal{F}^\lambda$ of feasible environment parameters, we use our knowledge of $\mathcal{F}^\lambda$ to enumerate a comprehensive set of feasible tasks on which every baseline's performance is measured. For the gridworld environment, the entire feasible set $\mathcal{F}^\lambda$ is calculated analytically. For MuJoCo, a discretization of the continuous parameter space $\Theta$ is calculated. Feasibility for each $\theta$ in the discretized space is then measured by training an RL best-response $\mathbb{BR}(\theta)$ to completion and averaging final utility $U_p(\mathbb{BR}(\theta), \theta)$ over 7 seeds. $\mathcal{F}^\lambda$ is then determined using equation (1). We measure feasible worst-case episode reward as $\min_{\theta \in \mathcal{F}^\lambda} U_p(\sigma_p, \theta)$, averaging over 100 episodes for each value of $\theta$. We use this as our metric for robustness to the feasible set $\mathcal{F}^\lambda$.

## 5.1 LAVA WORLD

The gridworld task "Lava World" consists of a small platform surrounded by lava, as depicted in Figure 2 (a). The adversary specifies a goal location $\theta$ that the protagonist needs to reach, however the protagonist does not observe the goal, and the goal can be placed in lava. With an episode horizon of 20, the protagonist receives a reward of -1 for every timestep that it does not reach the goal. If the protagonist moves into lava, the episode ends, and it receives a reward of -15 even if the goal is at that location. The feasibility threshold for this task is $\lambda = -10$, thus making a parameter for this environment infeasible if the goal is put in lava and feasible otherwise. We use DDQN (Van Hasselt et al., 2016) to train protagonist RL policies.

Worst-case protagonist episode reward among all values in the feasible set $\theta \in \mathcal{F}^\lambda$ is shown as a function of PSRO iterations for FARR and baseline objectives in Figure 2 (c). The minimax objective fails because the adversary learns to always suggest infeasible lava goals, and the protagonist learns to immediately jump in lava rather than waste time searching for a goal in non-lava cells. Likewise, domain-randomization fails because the majority of goals are infeasible lava goals, so in order to optimize the average-case, the protagonist learns the same suboptimal behavior as it does with minimax. The regret objective produces a distribution of both feasible and infeasible goals where non-lava goals are the majority, however because this distribution does not minimize protagonist reward over feasible goals, the regret protagonist does not maximize worst-case performance over the feasible set of tasks. FARR penalizes the adversary for suggesting the infeasible lava goals and otherwise provides base robust adversarial RL utilities for feasible goals, thus resulting in a protagonist that maximizes worst-case reward among the actual feasible non-lava goal set $\mathcal{F}^\lambda$.

## 5.2 MUJOCO

We compare FARR and other objectives on three MuJoCo control tasks, HalfCheetah, Walker2D, and Hopper using PPO (Schulman et al., 2017) to train protagonist RL policies. In each of these environments, the adversary specifies parameters $\theta = (\alpha, \beta)$ where $\alpha \in (0, 10], \beta \in (0, 10]$ for a beta distribution $\mathbf{B}(\alpha, \beta)$ used to sample from and generate 1D horizontal perturbing forces every

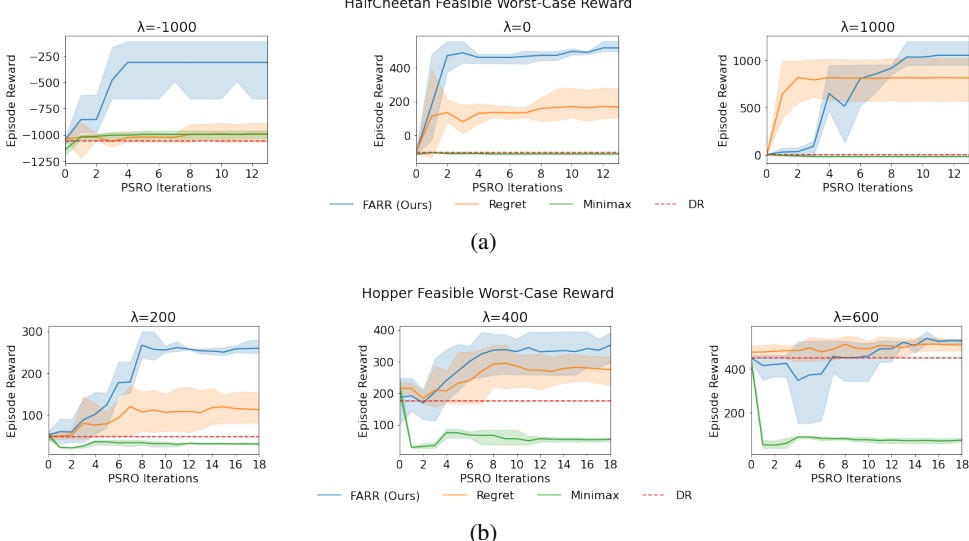

Figure 3: Worst-case MuJoCo HalfCheetah (a) and Hopper (b) average episode reward among task parameters in the feasible set $\mathcal{F}^\lambda$ as a function of PSRO iterations for FARR and other baselines with multiple values of $\lambda$.

timestep that are applied to the torso of the protagonist's robot. The adversary has the ability to specify infeasible distributions of forces which make accruing reward in each task virtually impossible. We conduct experiments with each MuJoCo environment using three different feasibility threshold values for $\lambda$, representing three different assumptions regarding the difficulty of test-time conditions that we wish to prepare for.

For HalfCheetah, in Figure 3 (a) we show worst-case protagonist reward among environment parameters in the feasible set $\mathcal{F}^\lambda$ as a function of PSRO iterations with $\lambda$ values $\{-1000, 0, 1000\}$. The same is shown for Hopper with $\lambda \in \{200, 400, 600\}$ in Figure 3 (b), and for Walker2D with $\lambda \in \{200, 400, 600\}$ in the appendix due to space limitations. Across different values for $\lambda$ in each of these environments, we see that FARR is able to train an agent which maximizes worst-case reward under the ground-truth $\mathcal{F}^\lambda$ by penalizing the adversary to prevent it from providing infeasible variations. Domain randomization and regret provide training distributions unconditioned on $\lambda$ or any notion of $\mathcal{F}^\lambda$, which are only sometimes appropriate for robust performance as seen for $\lambda = 600$ in Figure 3 (b) with domain randomization and regret and for $\lambda = 1000$ in Figure 3 (a) with regret. Otherwise, domain randomization and regret result in agents exploitable to some configuration in $\mathcal{F}^\lambda$. Likewise, minimax consistently provides insurmountable conditions to the protagonist, resulting in failed learning and highlighting the need for methods like FARR to automatically limit adversary abilities in robust RL. Analysis on the mixed strategies learned by each objective's adversary is available in Section C of the appendix.

## 6 DISCUSSION AND FUTURE WORK

We present FARR, a novel robust RL problem formulation and two-player zero-sum game objective in which we consider an underspecified environment allowing infeasible conditions and we train a protagonist to be robust only to the tasks which are feasible. By solving for approximate Nash equilibrium under the FARR objective using PSRO, we demonstrate that this method can produce a robust agent even when the adversary is allowed to specify parameters which make sufficient performance at a task impossible. A limitation and avenue for future work is that our current method for optimizing FARR does not directly optimize the adversary, instead relying on random search and the PSRO restricted game solution to provide an optimal mixed strategy. In future work, if high-dimensional joint adversary best-responses with feasibility estimates can be sample-efficiently optimized, FARR can provide a prescribable solution to avoid the manual creation of complex rules to limit robust RL adversaries in higher-dimensional sim-to-real configuration spaces.

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

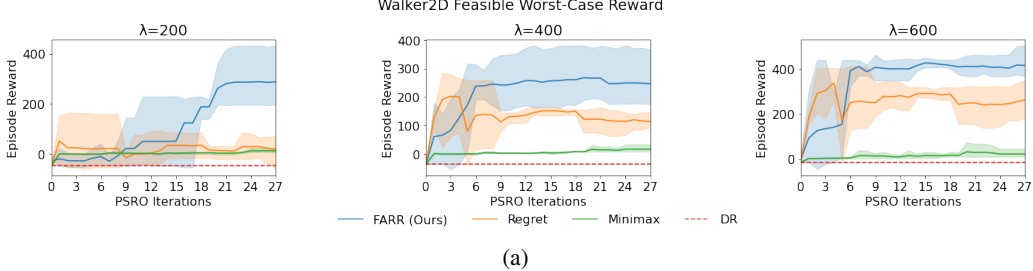

(a)

Figure 4: Worst-case MuJoCo Walker2D reward among task parameterizations in the feasible set $\mathcal{F}^\lambda$ as a function of PSRO iterations for FARR and other baselines with multiple values of $\lambda$.

## A    MUJOCO FEASIBLE SETS

For MuJoCo experiments, in order to measure each objective's worst-case average episode reward among feasible tasks, we evaluate on a discrete approximation of $\mathcal{F}^\lambda$. In these environments, $\Theta$ represents parameters of a beta distribution $\mathbf{B}(\alpha, \beta)$, $\alpha \in (0, 10], \beta \in (0, 10]$ sampled from each timestep to generate horizontal perturbing forces applied to the simulated robot. We consider a discretization of $\Theta$ with 11 different values in [0.01, 10] for both $\alpha$ and $\beta$. For each each combination $\theta = (\alpha, \beta)$, we train 7 seeds of a RL best-response $\mathbb{BR}(\theta)$ to completion using the same hyperparameters as the protagonist. The average final utility $U_p(\mathbb{BR}(\theta), \theta)$ across seeds is then used to calculate $\mathcal{F}^\lambda$ using equation (1). In Figure 5, for each environment, we show the 7-seed average $U_p(\mathbb{BR}(\theta), \theta)$ for every value of $\theta$ and the resulting feasible sets $\mathcal{F}^\lambda$ (shown in green) that we evaluate robustness to for each value of $\lambda$.

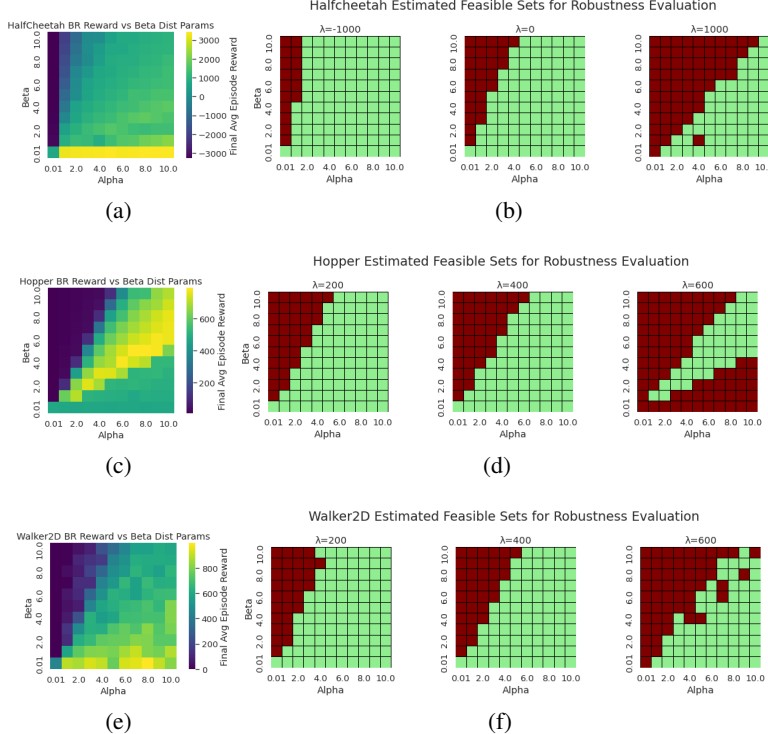

Figure 5: (a,c,e) Estimated values for $U_p(\mathbb{BR}(\theta), \theta)$ across the two parameters $\alpha$ and $\beta$ that the adversary has control over. (b,d,f) The feasible sets $\mathcal{F}^\lambda$ used for evaluation marked in green for each value of $\lambda$.

## B  MUJOCO LEARNED ADVERSARY STRATEGIES

After running PSRO to completion, the output strategies are both a protagonist mixed strategy $\sigma_p$ and an adversary mixed strategy $\sigma_\theta$ that should jointly approximate a Nash equilibrium to each of the two-player zero-sum game objectives we optimize. In figures 6, 7, and 8, we display the distribution of $\theta$ values induced by the final MuJoCo adversary mixed strategies $\sigma_\theta$ for the minimax, regret, and FARR objectives. For each $\lambda$ value considered, we overlay in shades of green the number of $\mathbb{BR}(\theta)$ seeds used in measuring $\mathcal{F}^\lambda$ that achieved a final average episode reward greater than or equal for $\lambda$.

Across all environments, the minimax adversary consistently selects the most difficult $\theta$ values possible outside of any variations considered feasible with the $\lambda$ values tested. In contrast, FARR mixes between $\theta$ values both well inside of the feasible regions and at the edges where task variations are as challenging as possible while remaining feasible.

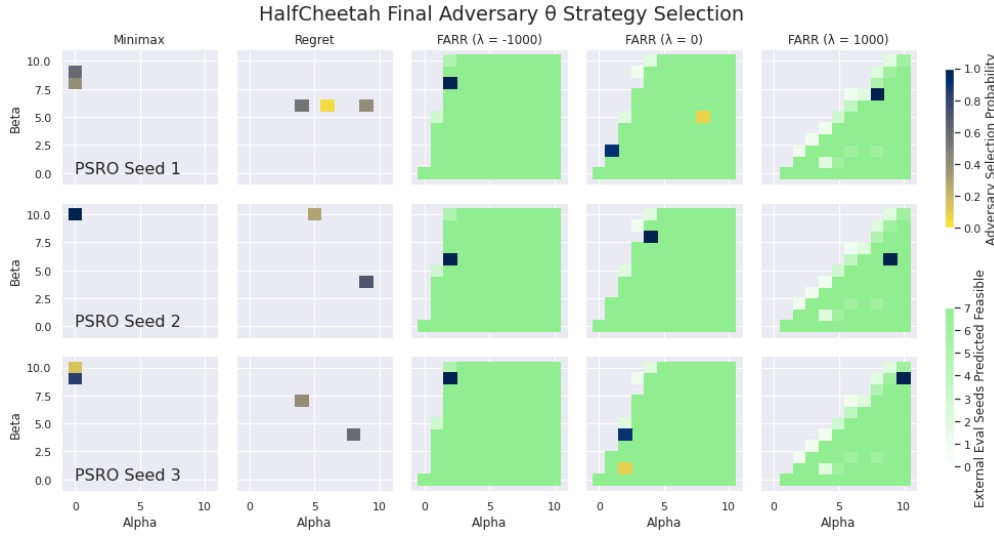

Figure 6: HalfCheetah $\theta$ distributions induced by the final adversary PSRO mixed strategy $\sigma_\theta$ for each objective.

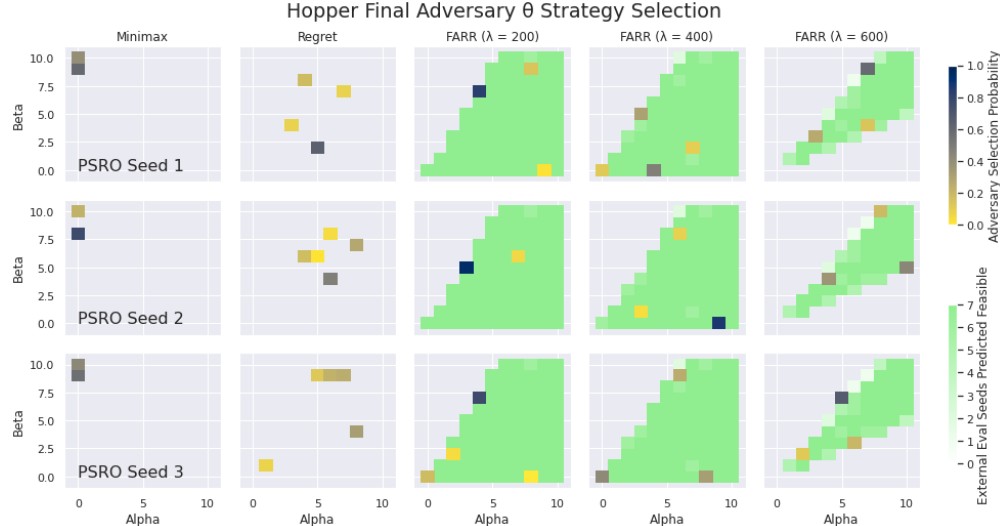

Figure 7: Hopper $\theta$ distributions induced by the final adversary PSRO mixed strategy $\sigma_\theta$ for each objective.

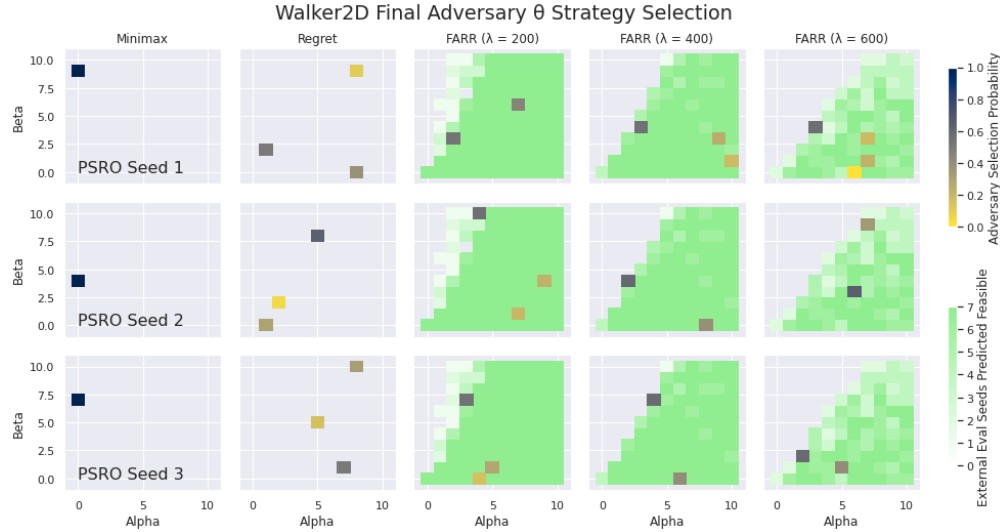

Figure 8: Walker2D $\theta$ distributions induced by the final adversary PSRO mixed strategy $\sigma_\theta$ for each objective.

## C  PROPERTIES OF NASH EQUILIBRIA FOR THE FARR TRANSFORMED GAME

In this section we show that solving for Nash equilibria in the FARR transformed game will give the same result as solving for NE in a regular minimax robust RL game with the adversary strategy space already limited to only feasible strategies. The benefit of solving the FARR game is that the set of feasible adversary strategies does not need to be known a priori.

Define the set of $\lambda$-infeasible adversary strategies as $\mathcal{I}^\lambda = \Theta \setminus \mathcal{F}^\lambda$. Define the set of all possible protagonist strategies $\pi_p$ as $\Pi_p^*$. For both protagonist utility functions $U_p^\lambda$ and $U_p$, the adversary's utility function $U_a^\lambda$ and $U_a$ is the negative of the protagonist's, $U_a^\lambda(\pi_p, \theta) = -U_p^\lambda(\pi_p, \theta)$ and $U_a(\pi_p, \theta) = -U_p(\pi_p, \theta)$ for any $\pi_p$ and $\theta$.

**Theorem 1.** *For sufficiently large C, a Nash equilibrium joint strategy $\sigma^*_{FARR}$ of a FARR transformed game $G_{FARR}$ with utility function $U^\lambda_p$, protagonist strategies $\Pi^*_p$ and all adversary strategies $\Theta$ is also a Nash equilibrium of a reduced game $G_{reduced}$ with utility function $U_p$, protagonist strategies $\Pi^*_p$, and only $\lambda$-feasible adversary strategies $\mathcal{F}^\lambda$.*

*Proof.* Let C take a sufficiently large value greater than any utility achievable by the protagonist $C > \max_{\pi_p, \theta} U_p(\pi_p, \theta)$. Since $U^\lambda_a(\pi_p, \theta') = -C$ when $\theta' \in \mathcal{I}^\lambda$, then for any $\theta' \in \mathcal{I}^\lambda$, any $\theta \in \mathcal{F}^\lambda$, and any $\pi_p \in \Pi^*_p$: $U^\lambda_a(\pi_p, \theta') < -U_p(\pi_p, \theta) = U^\lambda_a(\pi_p, \theta)$. It follows, as long as $\mathcal{F}^\lambda$ is nonempty, that for all $\theta' \in \mathcal{I}^\lambda$ there exists an adversary strategy $\theta \in \mathcal{F}^\lambda$ which achieves higher adversary utility against every $\pi_p \in \Pi^*_p$, thus all $\lambda$-infeasible adversary strategies $\theta' \in \mathcal{I}^\lambda$ are strictly dominated in the FARR transformed game.

If all $\theta' \in \mathcal{I}^\lambda$ are strictly dominated then they are not in the support of any Nash equilibrium for $G_{FARR}$. Furthermore, it is possible to remove strategies $\theta' \in \mathcal{I}^\lambda$ though iterated elimination of strictly dominated strategies (IESDS) to reduce $G_{FARR}$ to $G_{reduced}$ since the adversary strategy set for $G_{reduced}$ is $\Theta \setminus \mathcal{I}^\lambda = \mathcal{F}^\lambda$ and for all $\theta \in \mathcal{F}^\lambda$ and $\pi_p \in \Pi^*_p$: $U^\lambda_p(\pi_p, \theta) = U_p(\pi_p, \theta)$. If $G_{reduced}$ is an outcome of IESDS from $G_{FARR}$, then if $\sigma^*_{FARR}$ is a NE of $G_{FARR}$, it is also an NE of $G_{reduced}$.

$\square$

By employing a penalty $C$ rather than directly pruning infeasible strategies from the PSRO restricted game, the FARR objective can be defined without consideration to the mechanics of any specific algorithm like PSRO. The FARR objective can potentially be optimized with two-player zero-sum game methods other than PSRO as well, though exploring the use of more optimization methods is left to future work.

## D   SELECTING VALUES FOR $\lambda$

FARR is most applicable when the appropriate value for $\lambda$ can be derived from problem requirements. For instance, $\lambda$ would ideally be set to the lowest average return that an optimal agent would receive across task variations in deployment. This value could come from the environment definition, where doable tasks provide a minimum level of return if accomplished. Alternatively, $\lambda$ could be chosen by anticipating the maximum difficulty of task variations that would be seen at test-time or on which robust performance is important to the practitioner.

In the case where an appropriate value of $\lambda$ is completely unknown and cannot be derived from problem requirements, in a low-dimensional task variation space, manually tuning the adversary's capabilities without FARR may be appropriate. In a complex, high-dimensional task variation space, searching for a useful $\lambda$ may be easier than a direct search over the space of adversary legal strategy sets because $\lambda$ presents a single variable to tune, rather than a large number of legal parameter ranges or complex conditional constraints between adversary-specified parameters that may need to be defined. In this work, we consider the case where $\lambda$ can be derived from problem requirements.

## E   PSRO COMPARISON WITH SELF-PLAY

In Lava World, for each of the two-player zero-sum game objectives, we compare PSRO to self-play in which the protagonist, adversary, and evaluator $\pi^\theta_e$ continuously train together. For all self-play agents, we train with PPO to enable stochastic policies like PSRO is able to output. Regret self-play matches the original PAIRED algorithm from Dennis et al. (2020).

Although self-play may potentially yield competitive performance in some scenarios, unlike PSRO, it lacks any guarantees of converging to an approximate Nash equilibrium in two-player zero-sum partially-observable Markov or extensive-form games. Seen in Figure 9, we see that self-play for both FARR and PAIRED fails to converge, reaching a maximum feasible-space worst-case average reward of -9 as agent policies cycle and learn to represent nearly deterministic strategies during most points in training. The NE for Lava World requires a mixed-strategy in which the adversary samples a high-entropy (non-uniform) distribution of hidden goals. The degenerate solution to Minimax is reached by both algorithms.

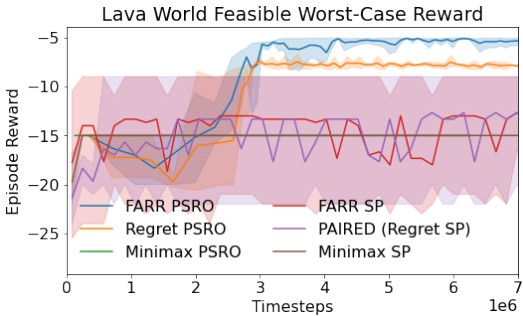

Figure 9: Worst-case average episode reward among goals in $\mathcal{F}^\lambda$ vs timesteps collected for each two-player zero-sum game objective optimized with both PSRO and PPO Self-Play.

## F  SAMPLE EFFICIENCY IN MUJOCO EXPERIMENTS

In figure 10, we show worst-case protagonist reward among environment parameters in the feasible set as a function of timesteps collected by PPO learners in PSRO for each $\lambda$ value considered. FARR and regret take more timesteps per PSRO iteration than minimax because they both train evaluation policies $\mathbb{BR}(\theta)$ for each adversary strategy $\theta$ added to the population in order to evaluate their utility functions in the PSRO restricted game.

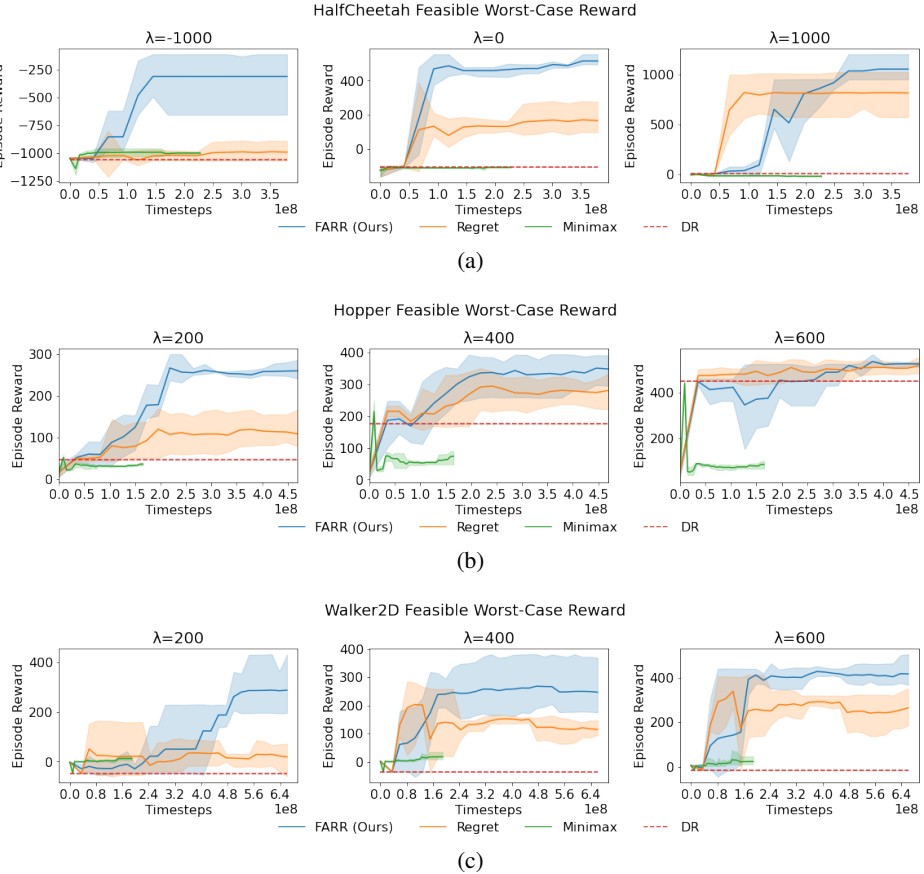

Figure 10: Worst-case MuJoCo HalfCheetah (a), Hopper (b), and Walker2D (c) average episode reward among task parameters in the feasible set $\mathcal{F}^\lambda$ as a function of timesteps collected for FARR and other baselines with multiple values of $\lambda$.

## G  ENVIRONMENT DETAILS

### G.1  LAVA WORLD

In the Lava World grid environment, the protagonist uses discrete actions to move in each of the 4 cardinal directions. For observations, the protagonist receives a one-hot encoding of its current location, and the protagonist does not observe the goal location. The adversary strategy space $\Theta$ is to define the hidden goal location and consists of every grid cell location in the environment's 5x5 grid with the exception of the protagonist's fixed starting location.

The protagonist receives a reward of -1 in each timestep that it does not reach the hidden goal location suggested by the adversary and -15 if it moves into a lava cell, even if the lava cell was a goal. An episode ends after either 20 timesteps elapse or the goal is reached.

### G.2  MUJOCO ENVIRONMENTS

The MuJoCo environments use the Mujoco physics engine (Todorov et al., 2012) and are modified versions of the perturbed robotic control environments originally presented in Pinto et al. (2017).

In each environment variation (HalfCheetah, Hopper, Walker2D), a max episode duration of 200 timesteps is imposed, and the proportion of time remaining in the range [0, 1] is appended to each task's original observation. We use the continuous action space variants of each task.

The adversary strategy space $\Theta$ consists of continuous $\alpha$ and $\beta$ parameters in the range (0, 10] for a beta distribution $\mathbf{B}(\alpha, \beta)$ used to generate horizontal perturbing forces sampled and applied to the robot's torso every timestep. Each timestep, a new horizontal force $F \in [-F_{\max}, F_{\max}]$, $F = X(2F_{\max}) - F_{\max}$ is generated where $X \sim \mathbf{B}(\alpha, \beta)$ and $F_{\max} = 100$.

When discretizing values of $\theta = (\alpha, \beta)$ for evaluation purposes to measure a policy's performance across values in $\Theta$ or $\mathcal{F}^\lambda$, we use $\alpha, \beta \in \{0.01, 1.0, 2.0, 3.0, 4.0, 5.0, 6.0, 7.0, 8.0, 9.0, 10.0\}$.

## H  TRAINING DETAILS

Protagonist RL training details are provided below for each environment. Policies used to estimate $\mathbb{BR}(\theta)$ for a given $\theta$ use the same training procedure and parameters as the protagonist. Like Dennis et al. (2020), we train protagonist policies on the easier-to-learn unmodified environment reward $U_p$ rather than our two-player game objective $U_p^\lambda$ because the only component of the game utility that the protagonist can affect is $U_p$. A protagonist best-response that maximizes $U_p$ also maximizes $U_p^\lambda$. Critically, we still calculate $U_p^\lambda$ in the PSRO empirical payoff matrix $U_\lambda^\Pi$ and use $U_\lambda^\Pi$ to calculate the meta-game NE strategy $\sigma = (\sigma_p, \sigma_\theta)$.

### H.1  LAVA WORLD

We train Lava World protagonist RL policies using DDQN Van Hasselt et al. (2016). All Lava World protagonist RL policies are stopped training after either 150,000 timesteps are collected or once performance plateaus (average episode return doesn't improve by 0.5 over 20,000 timesteps and a minimum of 80,000 timesteps is collected). Lava World DDQN hyperparameters are presented below. Our RL code was built using the RLlib framework Liang et al. (2018), and any hyperparameters not specified are the version 1.0.1 defaults. We use an infeasibility penalty of $C = 50$.

| algorithm | DDQN Van Hasselt et al. (2016) |
|---|---|
| circular replay buffer size | 50,000 |
| prioritized experience replay | No |
| total rollout experience gathered each iter | 8 steps |
| learning rate | 0.007 |
| batch size | 1024 |
| optimizer | Adam (Kingma & Ba, 2014) |
| TD-error loss type | MSE |
| target network update frequency | every 4,000 steps |
| MLP layer sizes | [256, 256] |
| activation function | tanh |
| discount factor $\gamma$ | 1.0 |
| exploration $\epsilon$ | Linearly annealed from 0.5 to 0.01 over 20,000 timesteps |

Table 1: Lava World protagonist DDQN hyperparameters

| algorithm | PPO Schulman et al. (2017) |
|---|---|
| GAE $\lambda$ | 0.9 |
| entropy coeff | 0.007 |
| clip param | 0.276 |
| KL target | 3e-4 |
| KL coeff | 0.0016 |
| learning rate | 5e-4 |
| train batch size | 8192 |
| SGD minibatch size | 64 |
| num SGD epochs on each train batch | 40 |
| shared policy and value networks | No |
| value function clip param | 10 |
| MLP layer sizes | [256, 256] |
| activation function | Tanh |
| discount factor $\gamma$ | 1.0 |

Table 2: Lava World PPO self-play protagonist hyperparameters

| algorithm | PPO Schulman et al. (2017) |
|---|---|
| GAE $\lambda$ | 0.95 |
| entropy coeff | 0.006 |
| clip param | 0.292 |
| KL target | 0.092 |
| KL coeff | 0.168 |
| learning rate | 3e-4 |
| train batch size | 8192 |
| SGD minibatch size | 64 |
| num SGD epochs on each train batch | 30 |
| shared policy and value networks | No |
| value function clip param | 100 |
| MLP layer sizes | [256, 256] |
| activation function | Tanh |
| discount factor $\gamma$ | 1.0 |

Table 3: Lava World PPO self-play adversary hyperparameters

In PSRO, to calculate the payoff matrix $U_\lambda^\Pi$, we estimate $U_p(\pi_p, \theta)$ for each pairing of player policies $\pi_p \in \Pi_p$ and $\theta \in \Pi_\theta$ using a single rollout because both Lava World environment dynamics and evaluation DDQN policies are deterministic. The normal-form meta-game Nash Equilibrium over $U_\lambda^\Pi$ is calculated using 2000 iterations of Fictitious Play (Brown, 1951). Calculating the meta-game NE typically takes a second or less of wall-time compute.

During PSRO evaluation, we measure the performance of the protagonist meta-game mixed strategy $\sigma_p$, in which a new protagonist policy $\pi_p \sim \sigma_p$ is sampled at the begining of each episode.

In self-play, the adversary is trained as a single-step agent via PPO. For simplicity, we keep the same network architecture for all self-play agents, and the adversary observes a constant vector of zeros.

## H.2 MuJoCo

We train MuJoCo environment protagonist RL policies using PPO (Schulman et al., 2017). Each PPO model consists of an MLP followed by an LSTM with shared weights between the policy and value function, branching into final output layers after the LSTM. PPO hyperparameters for each MuJoCo environment are presented below. Any hyperparameters not specified are the RLlib version 1.0.1 defaults. We use an infeasibility penalty of $C = 1e6$.

| | |
|---|---|
| algorithm | PPO Schulman et al. (2017) |
| GAE $\lambda$ | 0.9 |
| entropy coeff | 0.01 |
| clip param | 0.001 |
| KL target | 0.004 |
| KL coeff | 0.522 |
| learning rate | 5e-4 |
| train batch size | 4096 |
| SGD minibatch size | 64 |
| num SGD epochs on each train batch | 5 |
| shared policy and value networks | Yes |
| value function loss coeff | 0.001 |
| value function clip param | 100 |
| continuous action range | [-1.0, 1.0] for each dim |
| MLP layer sizes | [32] |
| activation function | Tanh |
| LSTM cell size | 32 |
| LSTM max sequence length | 20 |
| discount factor $\gamma$ | 0.99 |
| RL policy training stopping condition | 7e6 timesteps |

Table 4: HalfCheetah PPO hyperparameters

| | |
|---|---|
| algorithm | PPO Schulman et al. (2017) |
| GAE $\lambda$ | 0.9 |
| entropy coeff | 0.001 |
| clip param | 0.002 |
| KL target | 0.036 |
| KL coeff | 0.013 |
| learning rate | 7e-4 |
| train batch size | 4096 |
| SGD minibatch size | 32 |
| num SGD epochs on each train batch | 5 |
| shared policy and value networks | Yes |
| value function loss coeff | 0.001 |
| value function clip param | 10 |
| continuous action range | [-1.0, 1.0] for each dim |
| MLP layer sizes | [64, 64] |
| activation function | Tanh |
| LSTM cell size | 32 |
| LSTM max sequence length | 20 |
| discount factor $\gamma$ | 0.99 |
| RL policy training stopping condition | 6e6 timesteps |

Table 5: Hopper PPO hyperparameters

| | |
|---|---|
| algorithm | PPO Schulman et al. (2017) |
| GAE $\lambda$ | 0.95 |
| entropy coeff | 0.0 |
| clip param | 0.014 |
| KL target | 0.005 |
| KL coeff | 0.007 |
| learning rate | 9e-4 |
| train batch size | 1024 |
| SGD minibatch size | 64 |
| num SGD epochs on each train batch | 10 |
| shared policy and value networks | Yes |
| value function loss coeff | 1e-4 |
| value function clip param | 1000 |
| continuous action range | [-1.0, 1.0] for each dim |
| MLP layer sizes | [32] |
| activation function | Tanh |
| LSTM cell size | 32 |
| LSTM max sequence length | 20 |
| discount factor $\gamma$ | 1.0 |
| RL policy training stopping condition | 6e6 timesteps |

Table 6: Walker2D PPO hyperparameters

In PSRO, to calculate the payoff matrix $U_\lambda^\Pi$, we estimate $U_p(\pi_p, \theta)$ for each pairing of player policies $\pi_p \in \Pi_p$ and $\theta \in \Pi_\theta$ using 100 rollouts as perturbed MuJoCo environment transition dynamics are stochastic. The normal-form meta-game Nash Equilibrium over $U_\lambda^\Pi$ is also calculated using 2000 iterations of Fictitious Play.

Although PSRO convergence guarantees are not provided for continuous-action environments, in McAleer et al. (2021), McAleer et al. (2022b), and our own experiments, PSRO reliably produces meta-game NE mixed strategies that are empirically difficult for an opponent to exploit.

## I    COMPUTATIONAL COSTS

Experiments were performed on a local computer with 128 logical CPU-cores, 4 RTX 3090 GPUs, and 512GB of RAM. Due to small network sizes and comparably high overhead of CPU-based environments, logging, and other tasks, most experiments were performed without GPU acceleration. All individual training runs for a given player against a fixed opponent took 5 CPU-cores each. Lava world experiments individually ran for roughly 8 to 24 hours each, while MuJoCo experiments individually ran for roughly 72 hours each.

## J    CODE

A GitHub link for our experiment code will be provided under the MIT license in an updated version of this work.

Our code is written on top of the RLlib framework (Liang et al., 2018) and uses environments built using the MuJoCo physics engine (Todorov et al., 2012), both of which are open-source and available under the Apache-2.0 Licence.

