# OpenReview forum: "Feasible Adversarial Robust Reinforcement Learning for Underspecified Environments"
_ICLR.cc/2023/Conference — Submitted to ICLR 2023_

### Official Review · Reviewer_yKnP · 2022-10-24

**Confidence:** 5
**Correctness:** 3
**Technical Novelty And Significance:** 1
**Empirical Novelty And Significance:** 1
**Recommendation:** 3

**Clarity, Quality, Novelty And Reproducibility:**

Clarity: good.
Quality: fair.
Novelty: low.
Reproducibilty: high.

**Strength And Weaknesses:**

Strengths:
1. The problem of robust optimization with feasible parameter set is interesting and practical. The setting is well-motivated.
2. The line of related work seems to be well-discussed.

Weaknesses:
1. Although the problem is well-motivated, the formulation is somewhat trivial since it requires prior-knowledge of the feasible set of parameters. This is either directly fed as prior-knowledge, or using the penalty coefficeint C (which itself is unknown and questionable) as well as the threshold $\lambda$ for determining the feasible set. This makes the practicality of the algorithm very limited. In some sence, if one treats it as prior knowledge, the problem reduces to the same standard robust RL problem where the adversary's feasible set of parameters is specified.
2. Because the essense of the formulation is still aligned with a typical robust RL setting, the algorithms for robust RL, like PSRO, are still applicable to the FARR formulation. And the methodological/technical contribution is therefore very limited.
3. In evaluation, the solution based on the FARR formulation is better than the baselines. This is not surprising since the baselines are not fed with the feasibility information that the FARR formulation has.
4. The feasiblity definition is a bit contrived in the sense that it essentially removes all the environments where the expected return for the protagonis agent is small (than a threshold). First of all, the threshold is not always known, or sometimes unclear. Second, it is not equivalent to "feasible". In practice, many feasible parameters may yield very low return, but you cannot just ignore these parameter.

**Summary Of The Paper:**

The paper studies the robust reinforcement learning problem where the set of feasible environmental parameters are restricted. The authors propose the Feasible Adversarial Robust RL (FARR) formulation to this problem which specifies that the feasible parameter set is the set where the expected return of the protagonist agent is not less than a certain threshold. Essentially being a min-max game, the authors then solve the FARR problem using the Policy Search Response Oracle (PSRO) algorithm.

**Summary Of The Review:**

My main concerns are summarized at the weaknesses part of the above.

---

> ### Author Response · Authors · 2022-11-18
> **Author Response to yKnP**
>
>
> The authors thank the reviewer for the thoughtful feedback.
>
> [1] We do not provide the feasible set of parameters as prior knowledge. This is learned and approximated at training time. FARR is designed for situations in which practitioners are able to anticipate the maximum difficulty among tasks that are expected at test-time, over which robustness is desirable. This maximum difficulty is provided at training time as the scalar value $\lambda$. This is the only information that we provide to FARR other than $C$, a penalty used in optimizing the FARR objective, which simply needs to be an arbitrarily large positive value (We describe the sufficient magnitude for $C$ in the appendix, section C).
>
> We have clarified these points in a revision of the paper.
>
> [2] FARR was purposefully designed to be a stand-in swappable objective for existing optimization methods like PSRO. We argue that this is a strength of our approach.
>
> [3] We are empirically demonstrating: A) our ability to effectively train an optimal solution for the FARR problem formulation and B) the pitfalls of using other objectives when an environment is sufficiently underspecified and allows undesirably-difficult task variations. We agree that, given the ability to effectively optimize the intended FARR objective, the relative performance between it and other reasonable objectives is not surprising.
>
> [4] In this context, can you please provide your definition of "feasibility"? In our work, per equation (1), we define "feasible" tasks explicitly as those on which an optimal policy could achieve at least $\lambda$ average return. If there are tasks on which one cares about performance where $\lambda$ return is not possible, $\lambda$ should be lowered to meet that expectation. FARR is most applicable when $\lambda$ can be inferred from problem requirements or if optimizing the scalar value $\lambda$ provides a lower dimensional search space than explicitly optimizing the space of legal task variations. We discuss how $\lambda$ values can be chosen in the appendix, Section D.

---

> > ### Comment · Reviewer_yKnP · 2022-12-09
> > **Not entirely convinced about feasibility**
> >
> > It looks like Reviewer Rk3h has the same concern regarding the feasibility information given to the algorithm. It is highly suggested that the authors clarify this.
> >
> > Except for that, the feasibility constraint $\lambda$ itself is also questionable. Though there is a discussion of how to obtain $\labmda$, it appears to be quite ad-hoc and heurstic-driven. And it's hard to have any guarantees, which can be dangerous to tasks that are, e.g., safety-critical which is a key objective in robust RL.

---

> > > ### Author Response · Authors · 2022-12-11
> > > **Author Response to yKnP**
> > >
> > > Can the reviewer please clarify which concern or aspect regarding the feasibility information given to the algorithm wasn't sufficiently addressed in our responses to [yKnP](https://openreview.net/forum?id=Su_HbZ0Sdz&noteId=jYcnbc31KXz) and [Rk3h](https://openreview.net/forum?id=Su_HbZ0Sdz&noteId=oWpbyLRMWY7) so that we may provide the necessary detail? Reviewer Rkwh seems to have [moved away](https://openreview.net/forum?id=Su_HbZ0Sdz&noteId=bd7BCHdJ1G) from their initial concerns following our reply.
> > >
> > > FARR provides a guarantee on the performance lower bound over the set of $\lambda$-feasible tasks. For both traditional minimax robust RL and FARR, a practitioner will be required to utilize domain knowledge to craft a suitable simulation and its allowed configuration space to ensure that the provided guarantee is useful for safe but not overly conservative performance in the actual deployment environment. FARR provides a possible route for the practitioner to better specify the allowed configuration space even when the best explicit definition of it cannot be given due to complex interactions. $\lambda$-feasibility is a natural enough concept that a practitioner can understand what tasks the agent will become optimally robust to for a given $\lambda$. They can choose a value in which all expected deployment conditions are in the $\lambda$-feasible set where performance guarantees are provided but unrealistically difficult conditions made possible by the simulator are excluded.

---

### Official Review · Reviewer_Rk3h · 2022-10-24

**Confidence:** 5
**Correctness:** 2
**Technical Novelty And Significance:** 1
**Empirical Novelty And Significance:** 1
**Recommendation:** 3

**Clarity, Quality, Novelty And Reproducibility:**

The novelty is that this applies a reward penalty for infeasibility rather than using regret which uses the performance of another agent (approximating the optimal policy).

**Strength And Weaknesses:**

### Strengths

* The idea is sensible and well motivated, this is clearly an important area for real-world RL and has received increased interest recently with methods such as PAIRED, PLR and ACCEL.
* The nice thing with this in principle is it allows the adversary to propose the hardest solvable environments, although that was not really demonstrated anywhere.


### Weaknesses

* The entire paper relies on an assumption that we have access to the feasibility of an environment, which is almost never the case in any of the practical settings where this may be used. The authors attempt to sweep this under the rug by saying "we note that, in order to evaluate our approach in this paper, we focus on environments where we can, in fact, calculate (the feasible region)". This is remarkable to me. The entire paper is focused on the idea of being able to learn a good policy with unfeasible regions where the feasibility is unknown, but then in the experiments we assume access to feasibility. This completely invalidates the entire point of the work, and in fact, makes me even more impressed by the performance of regret in the experiments which comes close despite not having extremely privileged information.
* The paper also implies that the reward penalty is fed back into a learned adversary which can effectively infer feasibility. Sadly this is also not true as the authors instead use random sampling.
* Given that the current method relies on random sampling, it should also compare against Robust PLR (Jiang et al, Replay Guided Adversarial Environment Design, NeurIPS 2021) which is a far stronger method empirically than PAIRED.
* The experiments are very toy, using HalfCheetah and Walker and then a tabular MDP. The field has moved on to much larger settings since the original RARL paper in 2017. At minimum it should be considering more environments, something like the RWRL suite from pixels would make sense. This is not even super expensive to run.
* There is no analysis on the different types of emergent environments proposed by the different methods. This is crucial given toy experiments!
* What makes all of this even worse is that there is no limitations section where it is acknowledged that the paper doesn't do what it states out to do in the intro. If someone new to the field read this they may be led to believe FARR could infer the feasibility and that it was superior to regret based approaches. Please revise this before putting the paper on arxiv etc, even if unable to address the other issues.

**Summary Of The Paper:**

This work focuses on the robust reinforcement learning paradigm, whereby an agent is trained to be robust to worst case variations in an environment. The paper proposes an approach called FARR, whereby the adversary (proposing variations) is penalized if the environment proposed is not solvable. The paper sets out with great promise, working in an interesting area, but falls short with implementation details and experiments.

**Summary Of The Review:**

The idea makes sense and is well motivated. I was assuming from the intro that the authors had figured out how to learn to predict the feasibility of an environment, which seems like the hardest part of getting this to work. Unfortunately instead the method relies on knowledge of the feasibility which is not possible in any large scale environment we would care about. Further, baselines like regret perform well and don't need this.

Not only that, but FARR has no theoretical results (vs. regret-based approaches which do), and the experiments are low dimensional and toy, with no analysis on the different environments proposed by the adversaries. This paper makes sense as v0 of an idea for a workshop but is far below the bar for ICLR.

I would definitely provide a higher score if the authors figure out how to learn feasibility, provide greater analysis of the emergent environments proposed and also a greater breadth of tasks. This seems beyond the scope for a two week discussion phase but hopefully is possible for the next conference cycle.

---

> ### Author Response · Authors · 2022-11-18
> **Author Response to Rk3h**
>
> The authors thank the reviewer for the thoughtful feedback.
>
> [1] There seems to have been a misunderstanding with respect to the information provided to our agent at training time. We don't provide our agent the actual feasible region at training time, and we do learn an approximation of it as we claim. The only information we provide at training time is the scalar value $\lambda$, which indicates the target maximum per-task difficulty that we want to maintain, and the infeasibility penalty magnitude $C$, which is an arbitrarily large number.
> The sentence highlighted in your review, "we note that, in order to evaluate our approach in this paper, we focus on environments where we can, in fact, calculate (the feasible region)", only pertains to the test-time evaluation performed in the paper, and not to the algorithm itself.
> FARR-PSRO estimates feasibility by training a single best-response to each found adversary strategy $\mathbb{BR}(\theta)$ (analogous to the antagonist in PAIRED) and checking whether the average base-environment return is greater than or equal to $\lambda$. We have clarified these points in a revision of the paper.
>
> [2] The infeasibility reward penalty is provided to the adversary in the PSRO normal-form restricted game, where the FARR utility function is used to solve for the optimal mixed-strategy $\sigma_\theta$. $\sigma_\theta$ is the adversary component of the restricted game Nash Equilibrium, and as PSRO progresses, this closely approximates the Nash Equilibrium for the full game. Thus, we are able to use the FARR utility function with its penalty to find an optimal adversary despite the use of random sampling to add new adversary strategies to the PSRO population. In future work, we would like to investigate more efficient ways to guide our search for new adversary strategies, but the high-level method by which we find our optimal Nash Equilibrium adversary mixed-strategy would remain the same as the one demonstrated in this work.
>
> [3] Given that our comparisons are already between close approximations of each objective's Nash Equilibria found through PSRO, changing the speed at which we reach an optimal result with the regret objective would not change the overall properties of the optimal solutions we are comparing. The purpose of our experiments is not to compare one optimization method's ability to find an NE faster or in a larger class of games than another. We are demonstrating in our experiments that the optimal solution to the regret objective (which we are already finding a close approximation of in our experiments with PSRO) is not also the optimal solution to the FARR problem formulation despite partial overlap in motivations.
>
> [4] While we would always like to train on more environments, we believe that environments we do test in already sufficiently demonstrate A) our ability to effectively optimize the FARR objective and B) how other reasonable objectives can provide suboptimal solutions to the FARR problem formulation.
>
> [5] There is an analysis on the different types of emergent environments proposed by the different methods in the appendix, section B, titled "MuJoCo Learned Adversary Strategies". We will be sure to indicate this better in the main paper.
>
> [6] As discussed in response to point 1, we do learn feasibility and demonstrate what we state we do in the intro. To be clear, we do not claim that FARR provides a better solution to unsupervised environment discovery than regret, which works quite well. We argue that regret, though a reasonable approach for its own problem setting, is not also optimal for our adjacent considered problem of $\lambda$-feasible set robustness.
>
>
> We do provide theoretical results. These can be found in the appendix, section C, titled, "Properties of Nash Equilibria for the FARR Transformed Game".

---

> > ### Comment · Reviewer_Rk3h · 2022-11-23
> > **Not convinced**
> >
> > Hi, thank you for updating the paper and for clarifying some points.
> > Unfortunately I will not be changing my score. The problem setting seems contrived and it seems like you are comparing against just 1 actual baseline (Regret) in a very specific setting, providing your agent with $\lambda$ which requires domain knowledge. The experiments are toy, and do not appear to unlock some new "real world" capability. To me this does not meet the bar for publication because it seems overly specific without sufficient empirical evidence to motivate future work.

---

> > > ### Author Response · Authors · 2022-12-08
> > > **Author Response to Rk3h**
> > >
> > > Thanks for the response. Addressing each point:
> > >
> > > >The problem setting seems contrived
> > >
> > > This is a new criticism not previously made by the reviewer. Would the reviewer please clarify why they consider the problem setting contrived? It is challenging to apply minimax robust RL to configurable environments without carefully designing what the adversary is allowed to specify. We provide a way to apply the minimax robust RL solution concept while avoiding this issue.
> > >
> > > >  you are comparing against just 1 actual baseline (Regret)
> > >
> > > We compare with three alternative objectives, Minimax, Regret, and Domain Randomization because these are the only types of objectives that the community has been considering in problem settings that are adjacent to ours. Because this is a novel problem setting for which no existing method or objective was explicitly designed, we demonstrate that existing approaches are not sufficient for this and that a new approach (FARR) is needed to achieve an optimal solution.
> > >
> > > >  providing your agent with $\lambda$ which requires domain knowledge
> > >
> > > $\lambda$ is a scalar. Providing $\lambda$ does not require more than minimal domain knowledge (the scale of desired task difficulty). We argue that this is in most cases equal to or less than the amount of domain knowledge that is currently required to precisely tune the bounds of the minimax adversary variation space $\Theta$, as is needed for the traditional robust RL problem setting that we are extending.
> > >
> > > Our contribution is to show that even when complex interactions in a simulation make it difficult to program useful bounds to the adversary task variation space at the environment level, you can still effectively train a minimax robust RL agent by using our method and a single scalar piece of domain knowledge that implicitly defines which of the possible task variations you wish to be optimally robust to. FARR extends the class of training environments in which minimax robust RL methods can be applied to those where it is difficult to explicitly tune the adversary task space.
> > >
> > > >  The experiments are toy, and do not appear to unlock some new "real world" capability.
> > >
> > > Many important results in ML are initially demonstrated in modestly sized domains, such as MuJoCo, to ensure their viability before scaling up. This is particularly true in this work, as the paper explains, where finding the ground-truth $\lambda$-feasibility sets by brute force is necessary for the test-time evaluation of the method. The “real world” capability unlocked by this work is robust RL given poorly specified adversary task spaces — which has previously not been possible.

---

### Official Review · Reviewer_uRb1 · 2022-10-25

**Confidence:** 4
**Correctness:** 3
**Technical Novelty And Significance:** 2
**Empirical Novelty And Significance:** 3
**Recommendation:** 5

**Clarity, Quality, Novelty And Reproducibility:**

This paper is well-motivated and clearly written. The idea of using the threshold for the utility to define the uncertainty set is a natural approach, yet the approach is, to my knowledge, novel. However, it is largely based on the existing approach, PSRO, and its novelty is limited. The details of the experiments are given in the main part and in the appendix. Unfortunately, its code is not publicly available.

**Strength And Weaknesses:**

# Strength:

Approach: How to design the set of uncertainty parameters to be considered is a very important issue in robust reinforcement learning. Introducing a threshold for the utility and limiting the set of uncertainty to the parameters where the best possible performance is better than or equal to the threshold is a simple and natural approach. This paper proposes an approach to achieve this goal and show its usefulness in experiments.

Test Task Design: The test problem designed in this paper is, I think (not claimed by the authors), a good task revealing the issue of domain randomization, maximin approach, and regret minimization approach. It is very simple and easy to understand.

Evaluation: Numerical experiment on the test problem reveals the advantage of the proposed framework over domain randomization, worst-case maximization, and regret minimization.

 # Weaknesses
Min-Max vs Max-Min: In the proposed framework, the environment parameters where the optimal policy can reach the utility above the given threshold are considered as feasible parameters and the worst-case maximization is performed on such a restricted environment parameter set. This does not guarantee that the optimal policy for the worst-case in the restricted environment parameter set is above the threshold in general, because min-max is not equal to max-min. This limitation is not evaluated nor discussed in this paper.

Computational Time: In the experimental evaluation, the performance of the approach is displayed against PSRO iterations. Therefore, it is unclear how many interactions have been performed in the approach.

Evaluation (in particular on MuJoCo): The usefulness of the proposed approach is evaluated only on the basis of PSRO. However, because domain randomization, worst-case maximization, and regret minimization are proposed in conjunction with other baseline RL approaches, the usefulness of the proposed approaches over existing robust reinforcement learning approaches are not evaluated.

Regret minimization vs proposed approach: In Section 5.1, the authors say "The regret objective produces a distribution of both feasible and infeasible goals where non-lava goals are the majority, however this distribution is not an adversarial robust NE with respect to the set of feasible goals, so worst-case performance with the regret objective is still not optimal." It is understandable that the authors want to say that the optimal min-max regret solution is not the optimal min-max in the feasible environment. However, it is not clear why this is the case in this task. Please provide more explanation in relation with Theorem 1 or 2 of PAIRED paper. I feel that it is the matter of the magnitude of the reward. I guess min-max regret and the proposed formulation is the same if the penalty of lava is greater. At the same time, I understand that it is not always the case that there is a clear notion of success and we would like to have a good generalization ability in the set of environments where the optimal utility is sufficiently large. More explanation would be nice.


**Summary Of The Paper:**

This paper focuses on generalization performance in reinforcement learning: when applying the Robust Reinforcement Learning approach, it is important to know how to design the set of environmental parameters to be considered. The authors propose a method to approximate the Nash equilibrium solution in a zero-sum game, called Policy Space Response Oracles (PSRO), which is based on the Nash equilibrium solution in a zero-sum game. By applying a method for approximating Nash equilibrium solutions, called Policy Space Response Oracles (PSRO), a method is proposed for finding policies that maximize worst-case performance for a restricted set of environmental parameters. Experimental results show that higher utility can be achieved in a simple Grid World-like task and in a task using MuJoCo, compared to the case where Domain Randomization, Maximin, and Regret are used as measures.

**Summary Of The Review:**

Based on the above mentioned comment, I suggest weak accept. Although there are weak points, it has a potential to be usefulness in practice.

---

> ### Author Response · Authors · 2022-11-19
> **Author Response to uRb1**
>
> The authors thank the reviewer for the thoughtful feedback.
>
> **Min-Max vs Max-Min:** Because the FARR objective is a two-player zero-sum game, the min-max value does equal the max-min value. However  you do point out an important subtlety that we are interested in exploring further in future work. We argue that it makes sense to define feasibility on a per-task basis, as in eq. (1), because achievable task reward is a natural selection criteria for a user to specify over tasks that the protagonist might face in training. On the other hand, because the protagonist can have imperfect information regarding the task variation that it is experiencing, there is an inherent identification gap that may prevent the protagonist from achieving at least $\lambda$ return against the adversary Nash Equilibrium mixed strategy. Our per-task "feasibility" decision rule over task legality allows a practitioner to clearly understand and specify which tasks the final agent is optimally robust over.
> We will include this discussion in the final version.
>
>
>
> **Computational Time:**
> Figures for Lava-World with timesteps as the x-axis are included in the appendix, Section E. For MuJoCo, we have added figures with timesteps as the x-axis in the appendix, Section F. While minimax runs for at least as many PSRO iterations as FARR and Regret and converges to an approximate Nash Equilibrium, it requires less timesteps to do so because an evaluator policy $\mathbb{BR}(\theta)$ does not also need to be trained for each $\theta$ to evaluate the utility function like FARR and Regret do. We will include these graphs in the final version with minimax run for more PSRO iterations than other algorithms to fill the length of the graphs.
>
> **Regret minimization vs proposed approach:** In our problem formulation, the optimal solution is an adversary that provides a protagonist-reward-minimizing distribution over the space of feasible tasks and a protagonist that best-responds to this adversary mixed strategy, thus maximizing the protagonist's lower bound on feasible-task return. Because the protagonist has imperfect information w.r.t. the task variation it is experiencing, a protagonist best-responding to one distribution of tasks is not necessarily optimal against a different distribution with support in the same set of tasks.  If our protagonist best-responds to a different adversary mixed strategy than the one described above, it may not maximize its lower bound on feasible task return like we want.
>
> Regret, as described in theorem 2 of the PAIRED paper, is designed to provide a different solution concept than ours, providing an adversary that maximizes the difference in reward between a best response to each adversary strategy (the antagonist) and the protagonist. In relation to Theorem 1 in the PAIRED paper, even though in this lava-world case, $S_{max}-S_{min} \nless S_{min} - F_{max}$, the regret protagonist still happens to achieve $> S_{min}$ reward in feasible tasks in this example. However, because the protagonist has imperfect information and because regret provides a different distribution over tasks than a $\lambda$-feasible-minimax solution, the regret protagonist still doesn't maximize its lower bound on performance over feasible tasks. Instead, a regret NE selects a distribution of tasks that maximizes the difference between return for the antagonist that implicitly knows the task (because it's the same player as the adversary) and the protagonist, which doesn't know or directly observe the task.
>
>
> Likewise, even if we increase the difference between $F_{max}$ and $S_{min}$ (for instance, by increasing the jump-in-lava penalty), regret still isn't guaranteed to provide a strictly protagonist-reward-minimizing distribution over tasks with achievable rewards within $[S_{min}, S_{max}]$ like we want with the different problem formulation we consider in this work.

---

> > ### Comment · Reviewer_uRb1 · 2022-11-25
> > **Thank you for your response**
> >
> > Min-Max vs Max-Min: I am confused by the response to min-max vs max-min. Min-max = max-min is true for two-player zero-sum game with finite number of actions if mixed strategies are considered, thanks to Nash's existence theorem. However, it is not true in general if action is not in a finite set. In the context of the paper, this action corresponds to a possible policy, whose set is not even countable if I understand correctly. Why could you say min-max = max-min in the setting of the experiments using MuJoCo done in this paper for example?
> >
> > Computational Time: Thank you for providing the results. Timesteps of order 1e8 looks too much for these environments. Some related works in robust RL using MuJoCo requires two orders of magnitude smaller timesteps. To show the advantage of the proposed approach, I think you should add comparisons to other approaches that are not necessarily PSRO based.

---

> > > ### Author Response · Authors · 2022-12-08
> > > **Author Response to uRb1**
> > >
> > >
> > > Thanks for your response.
> > >
> > > **Min-Max vs Max-Min:**
> > > Yes, you're correct. Our "min-max=max-min" statement is with respect to finite games like our lava world example where Nash equilibrium convergence guarantees for PSRO can be applied. We don't make this claim with respect to the non-finite MuJoCo games in this paper. We note in Section H.2 of the appendix that Nash equilibrium convergence guarantees don't exist for the MuJoCo environments and that our results for those games are primarily empirical. We will include this in the main paper as well.
> > >
> > > The solution provided by PSRO when it is stopped will be a Nash equilibrium to the finite normal-form restricted game. The min-max value will equal the max-min value of the restricted game that PSRO solves for, but how closely the solution to this restricted game approximates a solution of the (possibly infinite) full game, in the case that such a solution exists, depends on the characteristics of the game and for how long PSRO is run. This consideration of how well a solution to the restricted game approximates one to the full game exists in a similar capacity for large finite games where PSRO will also need to be stopped early due to limited computation. In the control environments we consider, PSRO provides a low-exploitability solution in practice, but this will not always be the case, and careful consideration over the existence of a suitable approximate solution should be taken for any non-finite games. Likewise, if another two-player zero-sum algorithm solves a certain class of (approximately solvable) games better than PSRO, you could alternatively optimize the FARR objective with that method instead.
> > >
> > > **Computational Time:**
> > > We reemphasize that a contribution of this work is in proposing a novel objective that yields qualitatively different results when successfully optimized. We believe that this should be interesting to the community even before the question of how fast the objective can be optimized is fully explored.
> > >
> > > We also note that direct reference to the rough sample complexity of previous MuJoCo results is not clear-cut because we consider a new problem formation with an "underspecified" adversary variation space that is a superset of what would traditionally be used in a learnable robust RL setting. In MuJoCo, FARR converged on a timescale similar to Regret with a comparable PSRO optimization method, and in the Lava-World setting, PSRO outperforms optimization with self-play in the appendix, Section E.

---

### Official Review · Reviewer_9k7R · 2022-10-30

**Confidence:** 4
**Correctness:** 4
**Technical Novelty And Significance:** 4
**Empirical Novelty And Significance:** 2
**Recommendation:** 8

**Clarity, Quality, Novelty And Reproducibility:**

The manuscript has a great presentation and is enjoyable to read. The clarity and quality are excellent and the novelty is significant due to the introduction of the new robust RL setting. I did not check the reproducibility.

**Strength And Weaknesses:**

I agree that robust RL is indeed too conservative and the new feasible uncertainty set is relevant. The new setting proposed by the manuscript makes sense to me. It is also interesting that the setting could be solved by PSRO in quite an intuitive way. Some detailed reviews follow.

Pros:

1. The new setting is an interesting extension towards robust RL
2. The proposed algorithm through PSRO is intuitive and sensible.
3. Presentation of this manuscript is easy to follow.

Cons:

The major concern of mine is how useful this method could be, in practice. In the experiments, the agents that are trained against the *feasible adversary* demonstrates some decent performance in only 3 MuJoCo tasks (I have to assume that it doesn't pan out on the rest of the tasks). Notice that this is when the algorithm trained against the feasible adversary is tested against the feasible adversary. What if the agent trained against the feasible adversary is tested against a general adversary (potentially described by a real application task)? The manuscript does not provide such an answer, but I'm not very confidence on this.

Misc:

In the abstract "In real-world environments, choosing the set of possible values for robust RL can be a difficult task", the term "value" could be confused with the value function of RL. Use a different term.

**Summary Of The Paper:**

This manuscript investigates the setting where there exists a mismatch between the training MDP and the testing MDP (known as robust RL). While robust RL is quite conservative in that the adversary could minimize the protagonist's utility over every MDP in the uncertainty set, this manuscript believes that it is too conservative. Instead, the manuscript defined a feasible set of MDPs that the adversarial could achieve, by asserting that there must exist a policy in that adversarial MDP to achieve at least $\lambda$ in terms of the return. It subsequently lifts the constraint by replacing it with a substantially large penalty if the support of the random MDP of the adversarial has a positive density on the non-feasible area. With this setting, the robust RL policy and the adversarial could be trained by PSRO. Simulations are conducted on gridworld and 3 MuJoCo tasks.

**Summary Of The Review:**

I'm positive on this manuscript for its novelty in the new setting and its quality in presentation. I have reservation, though, on its empirical performance.

---

> ### Author Response · Authors · 2022-11-18
> **Author Response to 9k7R**
>
> The authors thank the reviewer for the thoughtful feedback.
>
> The reviewer brings up an important point. Just as the standard minimax Nash Equilibrium offers robustness guarantees only to the full set of training time environment variations, a FARR Nash Equilibrium offers robustness guarantees only to the feasible set of training time environment variations. Our tests are designed to empirically validate these claims, and performance outside these conditions is not guaranteed.
>
> FARR is most applicable when problem requirements can provide a suitable value for $\lambda$ a-priori so that our training time feasibility rule aligns with the difficulty of test-time “general adversary” conditions. When this isn't possible, $\lambda$ may alternatively serve as a hyperparameter that's lower dimensional than the explicit specification for the legal adversary strategy space, though we don't investigate this in our work. We discuss these points in Section D in the appendix.
>
> Our experiments were performed on a gridworld and 3 MuJoCo tasks. These tasks were chosen because we could objectively measure worst-case performance with respect to a true feasible set of tasks. No other environments were attempted for this work.
>
>
> Misc: Thanks, we've re-worded this in the latest version.

---

> > ### Comment · Reviewer_9k7R · 2022-12-02
> > **Thank you for your response**
> >
> > Thanks for the response. I would be curious why these 3 MuJoCo tasks are the only ones that you could measure the worst-case performance. Could you elaborate on this?

---

> > > ### Author Response · Authors · 2022-12-08
> > > **Author Response to 9k7R**
> > >
> > > Thanks for your question. We weren't strictly limited to the gridworld and 3 MuJoCo tasks, however to measure worst-case feasible-set performance in each of these, we did ensure that we could explicitly enumerate the feasible set $\mathcal{F}^\lambda$ for a given $\lambda$. With this computed "test-time" feasible set, we could measure the lowest average reward observed over all variations $\theta \in \mathcal{F}^\lambda$ for each objective/method.
> > >
> > > For lava world, we enumerated $\mathcal{F}^\lambda$ analytically.
> > >
> > > For MuJoCo, we enumerated a discretization of $\mathcal{F}^\lambda$ via brute force approximation by independently training best-response agents $\mathbb{BR}(\theta)$ to (121 = 11*11) different values for $\theta$. We filtered these $\theta$ values to enumerate our discretization of $\mathcal{F}^\lambda$ based on each best-response's final performance averaged over 7 seeds ($ \mathcal{F^\lambda} = \{ \theta \in \Theta | U_p(\mathbb{BR}(\theta), \theta)\ge\lambda \}$). The MuJoCo environments with an adversarial distribution of forces had the nice properties of a diverse enough variation space $\Theta$ to demonstrate the differences between solutions for various objectives while still allowing us to compute, for test-time evaluation and analysis, a good approximation of the entire feasible set $\mathcal{F}^\lambda$  in a tractable manner.

---

### Author Response · Authors · 2022-11-18
**Summary of Revision**

We thank the reviewers for their time spent reviewing and their feedback. We have submitted a revised version of our work with the following changes (highlighted in blue in the paper) and will continue to make refinements according to the reviewer discussions:

Main Paper:

- Better clarification in sections 4.2 and 4.5 on the data provided to FARR at training time, namely that only the scalar values $\lambda$ and $C$ are provided and that the actual feasible set is not provided to FARR at training time.

- Clarification in Section 2.3 on how regret's ability to produce a protagonist that achieves a "success" level of reward in a certain class of environments does not guarantee that it will learn optimal worst-case performance in a specific target set of tasks like is desirable for the FARR problem setting.

Appendix:

- Comparisons for MuJoCo experiments with timesteps as the x-axis added as Section F, "Sample Efficiency in MuJoCo Experiments".

---

### Decision · Program_Chairs · 2023-01-20

**Decision:**

Reject

**Justification For Why Not Higher Score:**

The reviewers pointed out several weaknesses in the paper and reached a rejection decision.

**Justification For Why Not Lower Score:**

N/A

**Metareview: Summary, Strengths And Weaknesses:**

The reviewers agreed that the paper considers an interesting setting for robust reinforcement learning and proposes a novel approach for defining uncertainty sets. However, the reviewers pointed out several weaknesses in the paper and shared common concerns. We want to thank the authors for their detailed responses. Based on the raised concerns and follow-up discussions, unfortunately, the final decision is a rejection. Nevertheless, the reviewers have provided detailed and constructive feedback. We hope the authors can incorporate this feedback when preparing future revisions of the paper.